# The Effect of Dual-Task Motor-Cognitive Training in Adults with Neurological Diseases Who Are at Risk of Falling

**DOI:** 10.3390/brainsci12091207

**Published:** 2022-09-07

**Authors:** Barbara Spanò, Massimo De Tollis, Sara Taglieri, Alice Manzo, Claudia Ricci, Maria G. Lombardi, Lorenzo Polidori, Ivo A. Griffini, Marta Aloisi, Vincenzo Vinicola, Rita Formisano, Carlo Caltagirone, Roberta Annicchiarico

**Affiliations:** Santa Lucia Foundation IRCCS, 00179 Rome, Italy

**Keywords:** fall, neurological disease, dual-task, motor, cognitive, gait, balance, fear of falling, walking speed

## Abstract

Falls are common in patients with neurological diseases and can be very problematic. Recently, there has been an increase in fall prevention research in people with neurological diseases; however, these studies are usually condition-specific (e.g., only MS, PD or stroke). Here, our aim was to evaluate and compare the efficacy of an advanced and innovative dual-task, motor-cognitive rehabilitation program in individuals with different neurological diseases who are at risk of falling. We recruited 95 consecutive adults with neurological diseases who are at risk of falling and divided them into four groups: 31 with cerebrovascular disease (CVD), 20 with Parkinson’s disease (PD), 23 with traumatic brain injury (TBI) and 21 with other neurological diseases (OND). Each patient completed a dual-task, motor-cognitive training program and underwent two test evaluations to assess balance, gait, fear of falling and walking performance at the pre-and post-intervention. We found that our experimental motor-cognitive, dual-task rehabilitation program was an effective method for improving walking balance, gait, walking endurance and speed, and fear of falling, and that it reduced the risk of falls in patients with different neurological diseases. This study presents an alternative approach for people with chronic neurological diseases and provides innovative data for managing this population.

## 1. Introduction

Falls are not only common in patients with neurological diseases, but are also very problematic. It is estimated that fall incidence is 2–4 times higher in patients with neurological disorders than in age-matched healthy subjects and that 46% of neurological patients fall at least once a year [1,2,3].

Fall-related injuries can also entail substantial medical costs and determine patients’ mortality risk [4].

Though not all falls are serious enough to require medical attention, it is known that all falls are predictors of future falls and can lead to a fear of falling. Falls and the fear of falling often cause a “post-fall syndrome”, i.e., a psychomotor regression condition responsible for psychological, postural and gait dysfunction [5,6,7,8,9].

The central nervous system oversees balance and gait control by integrating sensory inputs from the peripheral nervous system (e.g., receptors and nerves) and motor outputs to the musculoskeletal system [2,10,11,12]. Since cognitive-motor processes manage the spatiotemporal relationship between the body’s centre of mass and base of support, they are responsible for postural optimisation based on prior experience, current context and learning through long-latency components of postural responses [2,13]. In patients with impaired afferent sensory information, balance and gait control requires compensatory strategies such as attentional resources and sensory reweighting [2,14,15].

Patients with neurological disorders can manifest balance and gait dysfunction due to the impairment of at least one physiological component responsible for them. This significantly increases the risk of falls in this population compared to age-matched healthy subjects [1,2,16].

Studies that examined the risk of falling in neurological balance and gait disorders reported that the risk, frequency and consequences of falling in the presence of neurological gait disorders depend on whether the underlying disease entity is most pronounced in central rather than peripheral and functional etiologies [4,17]. Diseases for which an increased risk of falling has been documented include cerebrovascular disease (CVD), Parkinson’s disease (PD), multiple sclerosis (MS), dementia, cerebellar ataxia, traumatic brain injury (TBI), Huntington’s disease and peripheral neuropathy [1,2,3,4,16,18]. 

Due to the high incidence of falls and the associated negative consequences, preventing falls in people with neurological diseases is an important topic for research and the provision of healthcare services [19]. In recent years, there has been an increase in condition-specific fall prevention research in people with neurological diseases such as MS, PD and stroke. However, in the context of studies of the elderly, research concerning falls and neurological disorders is limited [19]. Furthermore, the implementation of single-diagnosis fall prevention interventions is challenging for the community and for primary care due to insufficient numbers of participants and resources for running separate group-based programmes [19]. Although there are differences in the underlying pathophysiology of these neurological conditions, research has identified many common fall risk factors in three conditions (i.e., MS, PD and stroke) [19]. The impairment of neurological functions that involve an increased risk of falling, irrespective of diagnosis, include disorders of balance and gait, lower extremity weakness or sensory loss, and loss of vision [2,3,18]. Given these similarities in fall risk factors across neurological diseases, the development of mixed-diagnosis interventions for these conditions can be a practical solution to bridge the intervention gap.

In this regard, several dual-task trainings have been shown to improve gait performance and to reduce the risk of falling in some neurological disorders, such as Parkinson’s disease and stroke [20,21,22,23,24]. Furthermore, in a recent pilot study [24] we demonstrated that our advanced and innovative dual-task, motor-cognitive training (DTT) is an effective method for improving walking balance, gait and walking speed, as well as reducing the fear of falling in patients with chronic CVD.

In light of these observations, the aim of this study was to explore and compare the effect of our DTT program [24] in adult patients at risk of falling with CVD, PD, TBI and other neurological conditions (OND).

## 2. Materials and Methods

### 2.1. Participants and Study Design

We recruited 95 adults with neurological diseases and at risk of falling who consecutively attended the Dual-Task Service at the IRCCS Santa Lucia Foundation. Patients were assigned to four groups according to their neurological disease: the cerebrovascular disease group (CVD, i.e., ischaemic and/or haemorrhagic stroke, *n =* 31), the Parkinson’s disease group (PD, *n =* 20), the traumatic brain injury group (TBI, *n =* 23) and the other neurological disease group (OND, i.e., multiple sclerosis, encephalitis, non-cancerous brain tumour, *n =* 21).

Inclusion criteria were the following: age ≥ 18 years; at least 5 years of formal education; risk of falling (total POMA, Tinetti Performance Oriented Mobility Assessment, score ≤ 20 and/or at least one fall in the previous year, in line with previous studies [24,25]).

Exclusion criteria were the following: presence of major cognitive and/or systemic disturbances; history of behavioural and/or psychiatric disturbances; receiving any kind of rehabilitative treatment.

### 2.2. Motor-Cognitive, Dual-Task Training (DTT)

All patients were trained during fifteen sessions of an individual, experimental dual-task rehabilitation program (i.e., simultaneous motor/cognitive tasks, 40 min/day, 3 days/week for 5 weeks). The intervention program was carried out in a dual-task room [24], i.e., the same as that previously used and described in detail in Spanò et al., 2022 [24]. Briefly, each DTT session consisted of the simultaneous administration of motor and cognitive tasks and included: (1) the first part of the protocol (i.e., 1/3 of the time of each training session) concerning the use of sensory carpets with different surfaces (medium density smooth, sandy and cobbled) and a video projector; (2) a second part of the protocol concerning the use of a walkable led floor and five video projectors (see Spanò et al., 2022 [24] for a detailed description).

To prevent errors, verbal instructions were provided by the therapist to assist patients in carrying out the exercises (e.g., “correct head position”, “correct feet position”, “correct balance”).

The DTT intervention was conducted in one-on-one training sessions that were adjusted daily according to the subject’s capability. Prior to the training, the therapists made a simple and rapid assessment of the patients in order to select the personalised intervention intensity suitable for them. Note that the programs can also be adjusted according to the patients’ own preferences. The intensity of each session was patient-specific, with rest breaks provided according to the therapist’s discretion and the patient’s tolerance of each activity. Each exercise provided increasing levels of difficulty, which were adjusted by the therapists based on the subject’s capability.

### 2.3. Outcomes

The outcome measure was the risk of falls evaluated with the following standardised scales [24]: (1) the Tinetti Performance Oriented Mobility Assessment (POMA) for balance (POMA-B) and gait (POMA-G) [26,27]; (2) the Falls Efficacy Scale-International (FES-I) [28,29]; (3) the six-minute walking test (6-MWT) [30,31] and gait speed (calculated as 6-MWT distance in metres divided by 360 s).

### 2.4. Data Analysis

Continuous variables are reported as median (interquartile range). Categorical variables are reported as frequency and percentage. At baseline, between-group differences in age, sex and clinical performances were tested using either the Kruskal–Wallis test or the χ^2^ test according to the level of measurement. 

Change values for the outcome measures were calculated by subtracting the baseline data from the post-intervention data. To analyse between-group improvement, the Kruskal–Wallis test was used.

The within-group effects (i.e., the difference in the outcomes observed between T0 and T1) were examined by adopting the Wilcoxon signed-rank test. The Z-score is also reported to represent the within-group effect size. For the outcomes, to avoid the type-I error, Bonferroni’s correction was applied (*p*-value threshold α = 0.05/6 = 0.008).

Statistical analyses were carried out using IBM SPSS, version 21.0 (SPSS Inc., Chicago, IL, USA).

## 3. Results

All recruited patients (31 CVD + 20 PD + 23 TBI + 21 OND) completed the DTT program and two outcome evaluations (one within 1 week before and one within 1 week after the intervention, i.e., 5 weeks later) and were included in the statistical analysis. During the study, no adverse events were encountered. Patients’ demographic and clinical results at baseline (T0) are summarised in Table 1. 

As expected, age was significantly different across groups: TBI patients were significantly younger than CVD and PD patients and OND patients were significantly younger than PD patients. Groups were significantly different also for sex distribution. Significant differences between groups were found for the POMA total score, POMA-B, and FES-I. However, no difference was found for the POMA-G, 6-MWT or gait speed.

Table 2 shows the measured outcomes at pre- (T0) and post-DTT (T1) and the change in values, expressed as a change between T0 and T1, and the between-group statistical results.

No significant between-group differences were found for any measured outcomes in the post-training change. 

In addition, post-DTT, the 6-MWT and gait speed were generally maintained in the TBI group, whereas improvements of 30 m or more and of 0.10 m/s or more, respectively, were found in the CVD, PD and OND groups (see Table 2).

Table 3 shows outcome results at pre- (T0) and post-DTT (T1) as well as the within-group differences. A significant improvement was found in all measures in both CVD and PD groups. In the TBI group, a significant improvement was found in the POMA total score, the POMA-B and POMA-G, but not in the FES-I and 6-MWT performance or in gait speed compared to pre-training. Then, in the OND group a significant improvement was observed post-DTT (T1) in the POMA total score, POMA-B and FES-I, but not in the POMA-G, the 6-MWT or gait speed. 

## 4. Discussion

The aim of the present study was to evaluate and compare the effectiveness of an experimental motor-cognitive DTT program in groups of patients with different neurological disorders (i.e., CVD, PD, TBI, OND) at risk of falls.

In a previous pilot study [24], we found that this DTT was a valid method for improving balance, gait and walking speed, as well as for reducing the fear of falling in elderly cerebrovascular patients, and that it is more effective than a sequential (mixed) motor-cognitive training. Here, we demonstrated that the same DTT is also feasible and effective in improving the physical performance of other neurological patients. 

The results of this study show that all groups (i.e., CVD, PD, TBI, OND) improved, but that the dual-task intervention was effective in all outcome measures (i.e., mobility and its balance and gait components, fear of falling, physical performance and gait speed) only in the CVD and PD groups.

As previously discussed in our pilot study [24], the slight superiority of the CVD and PD groups in terms of the outcomes obtained seems to be related to the specificity of the DTT because it simultaneously involves dual-task cognitive and motor aspects that are very important for these populations [23,32]. 

Although the DTT intervention did not statistically improve walking performance (6-MWT and/or gait speed) in the OND group, it is noteworthy that in this group walking endurance (6-MWT distance) and gait speed were improved by 33 m and 0.1 m/s, respectively; thus, the change was clinically meaningful [33]. Conversely, in the TBI group walking performance did not statistically improve after DTT, and walking endurance (6-MWT distance) and gait speed improved only by 18 m and 0.04 m/s. 

There are many factors that might explain these results.

First, the TBI group in the 6-MWT had a better performance at baseline than other groups, and thus, they may have had less room for improvement.

Second, the gait and balance deficits that characterise individuals with TBI vary considerably in terms of presentation pathophysiology [34] and severity [35,36]. This results in substantial variability in walking patterns among subjects, which may interfere with the average walking performance improvement.

Third, as recently shown by Acuña et al. [35], individuals who have experienced a prior TBI exhibit a decrease in neuromuscular complexity during gait. These data contribute to a growing body of evidence which suggests that brain injury reduces the complexity of the muscle activation patterns that underlie gait, and hence, may be due to a change in the use of cortical activity to modulate the rhythmic muscle activation patterns that underlie walking [35,37,38]. 

Finally, these results might reflect the characteristics of the specific population, which could have very different mechanisms underlying their impaired gait and motor control. Indeed, in TBI populations a relationship between complexity and walking performance or clinical assessments can also depend on the specific type of brain injury (e.g., a localised lesion within the brain) [39] or diffuse axonal injury [40]. 

Regarding the FES-I results, the DTT intervention was ineffective in reducing the fear of falling only in the TBI group. TBI survivors might present with low self-awareness [40,41,42] and a dysexecutive syndrome, which might also compromise the fear of falling estimate. Consistently with this speculation, it is also relevant that the patients with TBI obtained an initial average FES-I score of 23, which is considered an adequate score for classifying non-fallers [28].

Overall, the findings of this study show that the motor-cognitive, dual-task rehabilitation program we proposed is an effective and transversal treatment that is able to improve balance, gait and walking speed and to reduce the fear of falling in patients with different neurological conditions, such as CVD, PD, TBI and others who are at risk of falling. 

It is known that dual-task interference affecting walking performance has been observed in subjects with different neurological disorders. In recent years, there has been an increase in fall prevention research in people with neurological diseases; however, these studies are usually condition-specific (e.g., only MS, PD or stroke). To the best of our knowledge, no previous study has examined and compared the effects of a single dual-task, motor-cognitive training program in reducing the risk of falling in individuals with different neurological disorders (i.e., CVD, PD, TBI and OND). Thus, considering that falls and their prevention risk interventions are a major clinical problem in neurological patients, this study provides experimental support for future studies on implementing non-single-diagnosis fall prevention management. Moreover, this study demonstrates the importance of a motor-cognitive, dual-task intervention program to reduce the risk of falling in these patients.

This study has some important limitations. First, the age and sex between-group differences, the relatively small sample size of each group and the high clinical heterogeneity limit confidence in the effects that were observed and make it difficult to generalise the results. A larger, randomised, controlled clinical trial is needed to validate the benefits of the dual-task training protocols reported in the current study and to emphasise its neurological specificity. Second, as we did not perform a follow-up test, we are unable to evaluate the possible maintenance of the results over time. Third, as we did not make a neuropsychological assessment, we are unable to evaluate the effects of treatment on cognitive performance. Indeed, future research should consider this issue.

## 5. Conclusions

This study presents fall prevention management in patients with different neurological diseases who are at risk of falling. It demonstrates the importance of a motor-cognitive, dual-task intervention program to reduce the risk of falling in this population. The results suggest that our DTT has an adequate influence on the improvement of balance, gait, walking endurance and speed, and fear of falling in patients with different neurological diseases. Future studies should replicate this study so that its effects can be more confidently evaluated.

## Figures and Tables

**Table 1 brainsci-12-01207-t001:** Patients’ demographic and clinical results at baseline (T0).

	CVD (*n* = 31)	PD (*n* = 20)	TBI (*n* = 23)	OND (*n* = 21)	Between-Group Differences
Age (years) *	61.00 (23.00)	75.50 (15.50)	42.00 (32.00)	43.00 (25.50)	H = 38.84, *p* < 0.001 ^#, a,b,c,d^
Sex (male/female) **	20/11 (65/35)	5/15 (25/75)	21/2 (91/9)	9/12 (43/57)	χ^2^ = 21.92, *p* < 0.001 ^†, b,d^
POMA tot *	21.00 (6.00)	16.00 (6.75)	21.00 (5.00)	17.00 (6.00)	H = 17.17, *p* = 0.002 ^#, b,d^
POMA-B *	11.00 (4.00)	8.00 (3.75)	12.00 (3.00)	9.00 (5.00)	H = 16.62, *p* = 0.001 ^#, b,d,e^
POMA-G *	9.00 (2.00)	8.00 (2.00)	9.00 (3.00)	8.00 (2.50)	H = 7.80, *p* = 0.050 ^#^
FES-I *	25.00 (14.00)	32.00 (18.50)	20.00 (12.00)	28.00 (16.00)	H = 17.27, *p* = 0.001 ^#, e^
6-MWT (m) *	315.10 (158.00)	245.05 (264.13)	368.30 (163.00)	348.90 (188.05)	H = 6.17, *p* = 0.104 ^#^
Gait speed (m/s) *	0.90 (0.40)	0.70 (0.70)	1.00 (0.40)	1.00 (0.45)	H = 6.55, *p* = 0.088 ^#^

*Post hoc* comparison: ^a^ CVD versus TBI, *p*-value < 0.05; ^b^ PD versus TBI, *p*-value < 0.05; ^c^ PD versus OND, *p*-value < 0.05; ^d^ CVD versus PD, *p*-value < 0.05; ^e^ TBI versus OND, *p*-value < 0.05. CVD: cerebrovascular disease group; PD: Parkinson’s disease group; TBI: traumatic brain injury group; OND: other neurological disease group; POMA tot: Tinetti Performance Oriented Mobility Assessment total score; POMA-B: POMA balance score; POMA-G: POMA gait score; FES-I: Falls Efficacy Scale-International score; 6-MWT: 6 min walk test; m = metres; s = seconds. * Values are median (interquartile range); ** values are counts (percentage). ^#^ Kruskal–Wallis test. ^†^ Pearson’s χ^2^. See text for more details.

**Table 2 brainsci-12-01207-t002:** Value changes of the outcome measures at pre- and post-dual-task training and between-group statistical results.

	CVD (*n* = 31)		PD (*n* = 20)		TBI (*n* = 23)		OND (*n* = 21)		Between-Group Differences ^#^
	T0	T1	T0	T1	T0	T1	T0	T1	
**POMA-tot**	21.00 (6.00)	26.00 (4.00)	16.00 (6.75)	20.00 (5.75)	21.00 (5.00)	25.00 (5.00)	17.00 (6.00)	24.00 (5.50)	
T1–T0 change values	5.00 (3.00)		3.50 (3.00)		4.00 (3.00)		5.00 (3.50)		H = 3.90, *p* = 0.272
**POMA-B**	11.00 (4.00)	15.00 (3.00)	8.00 (3.75)	11.50 (3.75)	12.00 (3.00)	15.00 (3.00)	9.00 (5.00)	14.00 (3.00)	
T1–T0 change values	3.00 (3.00)		2.00 (2.75)		2.00 (2.00)		3.00 (2.50)		H = 8.22, *p* = 0.042
**POMA-G**	9.00 (2.00)	11.00 (3.00)	8.00 (2.00)	9.00 (2.75)	9.00 (3.00)	11.00 (3.00)	8.00 (2.50)	10.00 (3.50)	
T1–T0 change values	1.00 (1.00)		1.00 (2.00)		1.00 (1.00)		2.00 (2.00)		H = 0.91, *p* = 0.824
**FES-I**	25.00 (14.00)	20.00 (11.00)	32.00 (18.50)	31.45 (15.75)	20.00 (12.00)	22.00 (6.10)	28.00 (16.00)	24.00 (11.00)	
T1–T0 change values	−2.00 (7.00)		−3.50 (6.18)		0.00 (4.10)		−3.00 (6.50)		H = 11.32, *p*= 0.010
**6-MWT (m)**	315.10 (158.00)	358.30 (158.50)	245.05 (264.13)	259.55 (186.00)	368.30 (163.00)	408.00 (158.00)	348.90 (188.05)	390.00 (119.60)	
T1–T0 change values	32.00 (57.20)		43.75 (78.63)		42.30 (77.50)		20.00 (60.30)		H = 1.44, *p* = 0.696
**Gait speed (m/s)**	0.90 (0.40)	1.00 (0.50)	0.70 (0.70)	0.80 (0.55)	1.00 (0.40)	1.10 (0.50)	1.00 (0.45)	1.10 (0.35)	
T1–T0 change values	0.10 (0.20)		0.10 (0.20)		0.10 (0.20)		0.10 (0.15)		H = 1.54, *p* = 0.673

CVD: cerebrovascular disease group; PD: Parkinson’s disease group; TBI: traumatic brain injury group; OND: other neurological disease group; ^#^ Kruskal–Wallis test; POMA tot: Tinetti Performance Oriented Mobility Assessment total score; POMA-B: POMA balance score; POMA-G: POMA gait score; FES-I: Falls Efficacy Scale-International score; 6-MWT: 6 min walk test; m = metres; s = seconds. Values are expressed as medians (interquartile range). Change values were calculated by subtracting the pre-training (T0) data from the post-training (T1) data. See text for more details.

**Table 3 brainsci-12-01207-t003:** Outcome results at pre- and post-dual-task training and within-group statistical results.

	CVD (*n* = 31)		PD (*n* = 20)		TBI (*n* = 23)		OND (*n* = 21)	
	T0	T1	T0	T1	T0	T1	T0	T1
**POMA-tot**	21.00 (6.00)	26.00 (4.00)	16.00 (6.75)	20.00 (5.75)	21.00 (5.00)	25.00 (5.00)	17.00 (6.00)	24.00 (5.50)
Within-group differences ^#^	Z = −4.80, *p* < 0.001 *		Z = −3.76, *p* < 0.001 *		Z = −4.03, *p* < 0.001 *		Z = −3.93, *p* < 0.001 *	
**POMA-B**	11.00 (4.00)	15.00 (3.00)	8.00 (3.75)	11.50 (3.75)	12.00 (3.00)	15.00 (3.00)	9.00 (5.00)	14.00 (3.00)
Within-group differences ^#^	Z = −4.56, *p* < 0.001 *		Z = −3.67, *p* < 0.001 *		Z = −3.86, *p* < 0.001 *		Z = −3.94, *p* < 0.001 *	
**POMA-G**	9.00 (2.00)	11.00 (3.00)	8.00 (2.00)	9.00 (2.75)	9.00 (3.00)	11.00 (3.00)	8.00 (2.50)	10.00 (3.50)
Within-group differences ^#^	Z = −4.28, *p* < 0.001 *		Z = −3.37, *p* = 0.001 *		Z = −3.88, *p* < 0.001 *		Z = −2.61, *p* = 0.009	
**FES-I**	25.00 (14.00)	20.00 (11.00)	32.00 (18.50)	31.45 (15.75)	20.00 (12.00)	22.00 (6.10)	28.00 (16.00)	24.00 (11.00)
Within-group differences ^#^	Z = −3.26, *p* = 0.001 *		Z = −2.99, *p* = 0.003 *		Z = −0.10, *p* = 0.917		Z = −3.01, *p* = 0.003 *	
**6-MWT (m)**	315.10 (158.00)	358.30 (158.50)	245.05 (264.13)	259.55 (186.00)	368.30 (163.00)	408.00 (158.00)	348.90 (188.05)	390.00 (119.60)
Within-group differences ^#^	Z = −3.93, *p* < 0.001 *		Z = −3.06, *p* = 0.002 *		Z = −1.66, *p* = 0.10		Z = −2.05, *p* = 0.04	
**Gait speed (m/s)**	0.90 (0.40)	1.00 (0.50)	0.70 (0.70)	0.80 (0.55)	1.00 (0.40)	1.10 (0.50)	1.00 (0.45)	1.10 (0.35)
Within-group differences ^#^	Z = −3.45, *p* = 0.001 *		Z = −3.15, *p* = 0.002 *		Z = −1.45, *p* = 0.15		Z = −2.46, *p* = 0.01	

CVD: cerebrovascular disease group; PD: Parkinson’s disease group; TBI: traumatic brain injury group; OND: other neurological disease group; ^#^ Wilcoxon signed-rank test; * significant between-group difference at *p* < 0.008; POMA tot: Tinetti Performance Oriented Mobility Assessment total score; POMA-B: POMA balance score; POMA-G: POMA gait score; FES-I: Falls Efficacy Scale-International score; 6-MWT: 6 min walk test; m = metres; s = seconds. Outcome values are expressed as medians (interquartile range). See text for more details.

## Data Availability

The data presented in this study are available on request from the authors.

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
