# Peer review of "The Effect of Dual-Task Motor-Cognitive Training in Adults with Neurological Diseases Who Are at Risk of Falling"

_brainsci, 2022, doi:10.3390/brainsci12091207_

Round 1

Reviewer 1 Report

This manuscript requires some essential editing to improve the quality of the study.

Introduction: Paragraph 4- “The prevalence for gait impairments …………. peripheral neurological disorders.” Authors should discuss this statement more precisely that how the central neurological disorders increase the prevalence for gait impairments than peripheral neurological disorders. Also provide some reference to support such statement as the proposed dual task training is assessed mainly on central neurological disorders patients.

Methods: What sort of dual task training was provided to the patients?

Authors mentioned about the motor and cognitive tasks as Dual task training. Though there is no description about the training. Whether this motor-cognitive training was provided simultaneously or one after another in the same session or how rigorous was this training? Is there any resting interval between the 40 min session?

-What is the percentage of motor and cognitive task involved in this training. Is it same or more cognitive activity than motor or some other ratio?

Authors should provide more details about the dual task training as it is the essence of this study.

Author Response

Response to Reviewer 1 Comments

This manuscript requires some essential editing to improve the quality of the study.

The Authors wish to thank the Reviewer for his/her careful reading of the paper and for his/her constructive comments. We have addressed all Reviewer’s criticisms in the revised manuscript as detailed below. All changes in the manuscript are in blue characters and are marked up using the *Track Changes* function.

The revised manuscript was checked/edited by a native English-speaking colleague.

Introduction: Paragraph 4-“The prevalence for gait impairments …………. peripheral neurological disorders.” Authors should discuss this statement more precisely that how the central neurological disorders increase the prevalence for gait impairments than peripheral neurological disorders. Also provide some reference to support such statement as the proposed dual task training is assessed mainly on central neurological disorders patients.

We have now revised this paragraph and added some reference (please see page 1 and 2)

Methods: What sort of dual task training was provided to the patients?

Authors mentioned about the motor and cognitive tasks as Dual task training. Though there is no description about the training. Whether this motor-cognitive training was provided simultaneously or one after another in the same session or how rigorous was this training? Is there any resting interval between the 40 min session?

-What is the percentage of motor and cognitive task involved in this training. Is it same or more cognitive activity than motor or some other ratio?

The dual-task training consists of simultaneous administration of motor and cognitive tasks, stimulating patients to enhance both abilities, resulting in 40-min of training per session. Thus the percentage of motor and cognitive task involved in the training is the same. Intensity of each DTT session was patient-specific, with rest breaks provided upon therapist discretion and patients’ tolerance to activity.

We have now added this information in the text (please see page 3).

Authors should provide more details about the dual task training as it is the essence of this study.

In the first version of the paper we have presented a most detailed description of the Dual task training but we had to removed it upon request by Editors: the paper was too similar to our previously paper “Spanò, B.; Lombardi, M.G.; De Tollis, M.; Szczepanska, M.A.; Ricci, C.; Manzo, A.; Giuli, S.; Polidori, L.; Griffini, I.A.; Adria-no, F.; et al. Effect of Dual-Task Motor-Cognitive Training in Preventing Falls in Vulnerable Elderly Cerebrovascular Patients: A Pilot Study. Brain Sci. 2022, 12, 168. https://doi.org/10.3390/brainsci12020168“ in which we used the same Dual task training.

Therefore, in this new version of the paper we inserted the phrase “The intervention program, provided within a dual-task room [24], was the same previ-ously used and described in detail in Spanò et al. 2022 [24].” (please see page 3).

In Spanò et al. 2022 the dual task training description is very detailed (please see it).

Reviewer 2 Report

Although the aim of this study seems to be appropriate and interesting, there are major issues in the methodology that should be modified and reanalyzed. The detailed comments are as follows:

1) For the replication of the DDT program, the contents of the program should be presented in detail. Furthermore, authors are strongly recommended to present how to grade the level of difficulty of the program according to the subjects' responses or abilities.

2) In the baseline characteristics, ages and sex ratio significantly differ among the three groups. These differences should be considered in the between-group analysis as covariants. However, in the present form, this seems to be not implemented.

3) In table 2, between-group differences are indicated in the (POMA-B and FES-I). However, the authors reported that no significant between-groups differences were found for all measured outcomes. This disparity should be addressed. If there were significant differences as presented in table 2, post-hoc analysis should be conducted to indicate which groups differ.

4) The discussion section should be focused on what brings the differences in DDT's effect between groups. 

4) 

Author Response

Response to Reviewer 2 Comments

Although the aim of this study seems to be appropriate and interesting, there are major issues in the methodology that should be modified and reanalyzed. The detailed comments are as follows:

The Authors wish to thank the Reviewer for his/her careful reading of the paper and for his/her constructive comments. We have addressed all Reviewer’s criticisms in the revised manuscript as detailed below. All changes in the manuscript are in blue characters and are marked up using the *Track Changes* function.

The revised manuscript was checked/edited by a native English-speaking colleague.

1) For the replication of the DDT program, the contents of the program should be presented in detail.

In the first version of the paper we have presented a most detailed description of the Dual task training but we had to removed it upon request by Editors: the paper was too similar to our previously paper “Spanò, B.; Lombardi, M.G.; De Tollis, M.; Szczepanska, M.A.; Ricci, C.; Manzo, A.; Giuli, S.; Polidori, L.; Griffini, I.A.; Adria-no, F.; et al. Effect of Dual-Task Motor-Cognitive Training in Preventing Falls in Vulnerable Elderly Cerebrovascular Patients: A Pilot Study. Brain Sci. 2022, 12, 168. https://doi.org/10.3390/brainsci12020168“ in which we used the same Dual task training.

Therefore, in this new version of the paper we inserted the phrase “The intervention program, provided within a dual-task room [24], was the same previ-ously used and described in detail in Spanò et al. 2022 [24].” (please see page 3).

In Spanò et al. 2022 the dual task training description is very detailed (please see it).

Furthermore, authors are strongly recommended to present how to grade the level of difficulty of the program according to the subjects' responses or abilities.

The DTT intervention was conducted in one-on-one training sessions daily adjusted according to the subject’s capability. Before training, therapists make a simple and rapid assessment of patients to select the personalized intervention intensity suitable for patients. Programs can also be adjusted according to patients’ own preferences. The intensity of each session was patient-specific, with rest breaks provided upon therapist discretion and patients’ tolerance to activity. Each exercise provided increasing levels of difficulty adjusted by the therapists, consistent with the subject’s capability.

We have now added this information in the text (please see page 3).

2) In the baseline characteristics, ages and sex ratio significantly differ among the three groups. These differences should be considered in the between-group analysis as covariants. However, in the present form, this seems to be not implemented.

We concur with the Reviewer that sex and age should be considered in the between-group analysis as covariates. However, to the best of our knowledge, is not possible to controlling for a covariate using Kruskal-Wallis test and/or non-parametric statistics in SPSS. We insert this as a limit of the study (please see page 6).

Considering that no significant between groups differences were found for all measured outcomes’ post-training change, we can hypothesize that these variables did not influence the results (please see below)

3) In table 2, between-group differences are indicated in the (POMA-B and FES-I). However, the authors reported that no significant between-groups differences were found for all measured outcomes. This disparity should be addressed. If there were significant differences as presented in table 2, post-hoc analysis should be conducted to indicate which groups differ.

As expressed in “2.5 Data Analysis section” (please see page 3), for the outcomes, after the Bonferroni’s correction, the statistically significant threshold was set at p < 0.008. Thus, considering this threshold, between-groups differences are not significant for all measured outcomes’ post-training change, included POMA-B and FES-I.

4) The discussion section should be focused on what brings the differences in DDT's effect between groups. 

Please, see reply to point 3) above.

Reviewer 3 Report

The authors presented the paper with the aim to evaluate and compare the effectiveness of their motor-cognitive DTT program in groups of patients with different neurological disorders at risk of falls. The article is written well, and the English language is good. However, I have some questions and some remarks.

In the "2.1. Participants and Study Design" section, the authors used the acronym POMA without defining it first. I have noticed that it is defined later in the outcome measure, but I would like to see here also the full name of the test to make it more readable.

I also noticed that the authors used non-parametric Kruskal-Wallis and Wilcoxon signed-ranks tests. Is that because the data was not normally distributed? Also, the Wilcoxon test can be a good alternative to the t-test when population means are not of interest.

However, the authors decided to represent the mean and the standard deviation in the tables, which are used as parameters for the parametric test and do not well describe non normally distributed population. 

Therefore, I suggest reporting median + IQR (InterQuartile Range) when non-parametric tests are used, instead of mean+-stdev.

Author Response

Response to Reviewer 3 Comments

The authors presented the paper with the aim to evaluate and compare the effectiveness of their motor-cognitive DTT program in groups of patients with different neurological disorders at risk of falls. The article is written well, and the English language is good. However, I have some questions and some remarks.

We thank Reviewer for his/her positive comments. The Reviewer also raised some issues that we have addressed in the revised manuscript as detailed below. All changes in the manuscript are in blue characters and are marked up using the *Track Changes* function.

The revised manuscript was checked/edited by a native English-speaking colleague.

In the "2.1. Participants and Study Design" section, the authors used the acronym POMA without defining it first. I have noticed that it is defined later in the outcome measure, but I would like to see here also the full name of the test to make it more readable.

The Reviewer is right. These mistakes have been corrected in the revised manuscript (please see page 3)

I also noticed that the authors used non-parametric Kruskal-Wallis and Wilcoxon signed-ranks tests. Is that because the data was not normally distributed? Also, the Wilcoxon test can be a good alternative to the t-test when population means are not of interest.

Yes, our data was not all normally distributed at the Shapiro-Wilks test. Furthermore, our data were not all continuous variables and patients’ subgroups were small. Thus, to the best of our knowledge, non-parametric statistical tests are more appropriate than parametric statistical tests.

In the statistical analysis we used the Kruskal-Wallis test to analyse between-group improvement and the Wilcoxon signed rank test to evaluate within-group effects. To the best of our knowledge, these different statistical test are very appropriate for our different measure design (between group and within group).

Wilcoxon rank-sum test is used to compare two independent sample, while Kruskal-Wallis three or more unpaired groups.

Therefore, I suggest reporting median + IQR (InterQuartile Range) when non-parametric tests are used, instead of mean+-stdev.

We have modified it in the resubmitted manuscript (please see Table 1, Table 2 and Table 3).

The Wilcoxon signed rank test is for related or matched samples (like 'repeated measures' on the same subjects). It is very appropriate for a repeated measure design where the same subjects are evaluated under two different conditions such as pre- post-treatment.

However, the authors decided to represent the mean and the standard deviation in the tables, which are used as parameters for the parametric test and do not well describe non normally distributed population.

Round 2

Reviewer 2 Report

The issues I suggested are all well addressed. There is no further revision to request.